# LLM Self-Recognition:
# Steering and Retrieving Activation Signatures

**Thibaud Ardoin** [1]   **Jonas Schäfer** [1]   **Gerhard Wunder** [1]

## Abstract

Recent advances in interpretability suggest that large language models (LLMs) implicitly encode signals in their generated text that enable self-recognition of their outputs. We demonstrate that this capability is reliable, even in low-entropy scenarios, and that it can be amplified through targeted intervention. By steering the internal residual stream during generation with a random sparse vector, we create a detectable fingerprint that enables attribution of a given text to a specific LLM. This signal is recoverable from the activations of an LLM used as a detector, achieving over $98\%$ accuracy across multiple detection settings while preserving the quality of generated text. As AI-generated content proliferates, this approach offers a practical alternative to traditional detectors by leveraging the model's natural representation structure for attribution rather than embedding a signal externally. Our contributions include: (i) establishing reliable self-recognition capabilities in LLMs, (ii) a simple steering mechanism enabling multi-LLM identification with no quality degradation, (iii) demonstrating that activation spaces contain exploitable structure for encoding signals without semantic interference.

## 1. Introduction

As large language models (LLMs) become increasingly deployed for content generation, concerns about the authenticity and traceability of AI-generated text have intensified. Unlike human-authored text, LLM generations often lack an intrinsic link to a responsible author, raising risks of knowledge base contamination and accountability gaps in

(a) Steering at generation to create a unique signature

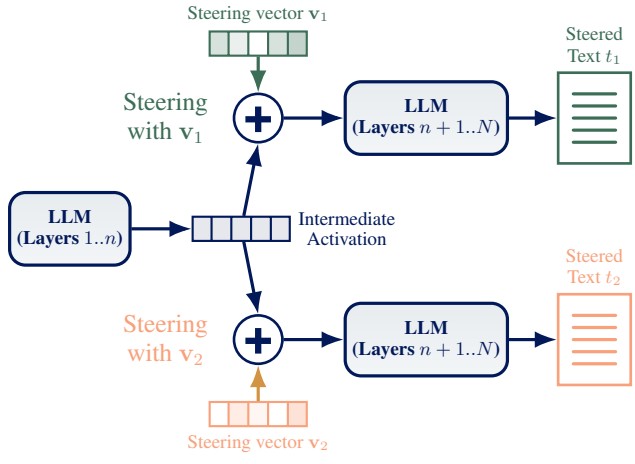

(b) Extracting activations from the text
and retrieving the steering vector

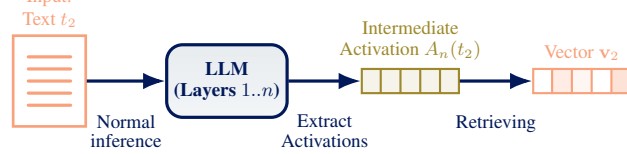

*Figure 1.* Steering and retrieval of LLM signatures. (a) During generation, steering vectors $\mathbf{v}_1$ and $\mathbf{v}_2$ are added to intermediate activations at layer $n$, creating model-specific signatures in outputs $t_1$ and $t_2$. (b) To verify authorship, the generated text is passed through the same model to collect the activations at layer $n$. The source steering vector is retrieved via cosine similarity or a trained MLP classifier.

high-stakes domains. Distinguishing not only whether content is AI-generated, but which specific model produced it, is essential for auditing, attribution, and misuse prevention (Bjelobaba et al., 2025; Yousaf, 2025).

Current approaches to AI-generated text detection (AI-GTD) predominantly rely on watermarking methods that either modify token probability distributions (Kirchenbauer et al., 2023) or strategically select tokens to encode hidden information (Hou et al., 2023; Christ et al., 2024). Integrating these methods into the generation process incurs additional

[1]Department of Computer Science, Freie Universitaet Berlin, Germany. Correspondence to: Thibaud Ardoin <thibaud.ardoin@fu-berlin.de>.

*Proceedings of the 43rd International Conference on Machine Learning*, Seoul, South Korea. PMLR 306, 2026. Copyright 2026 by the author(s).

overhead, and potential trade-offs between detection robustness and output quality have limited their widespread adoption (Sadasivan et al., 2023). Alternatively, AI-GTD can be approached post-generation using classifiers that exploit statistical properties of generated text (Mitchell et al., 2023) or classifiers trained on large labeled corpora. However, these detection methods are sensitive to distribution shifts across domains and do not naturally extend to distinguishing among multiple LLMs.

While our approach targets a different set of trade-offs than adversarially robust watermarking, it offers a fundamentally different lens on model attribution. Rather than relying on external watermarking mechanisms applied at the output token level, we investigate whether a signature can be embedded and recovered by exploiting the structure of the model's internal representations. Specifically, we ask whether models naturally encode a fingerprint, how such signals can be detected, and whether an additional, easily recoverable signature can be embedded alongside them.

Recent work in interpretability has revealed that LLMs can indeed recognize their own generated outputs with nontrivial accuracy (Ackerman & Panickssery, 2025; Bowman et al., 2024). This capability, termed *self-recognition*, suggests that models implicitly encode model-specific information in their generations. At the same time, activation engineering has demonstrated that targeted interventions in internal activations can steer LLM behavior while largely preserving output quality (Panickssery et al., 2023; Liu et al., 2023). Together, these findings suggest that internal activations offer a principled mechanism for detecting and inducing model-specific generation signatures.

Building on these advances, we propose to use the internal representations of an LLM as the space in which authorship signals are detected. Furthermore, we show that inference-time steering can be used to deliberately inject distinct and recoverable signatures, enabling more reliable detection and supporting multi-model attribution without sacrificing generation quality. Our approach enables a flexible detection framework, covering a range of scenarios, from post-generation AI-GTD to the identification of multiple versions of the same model through steering. We release code and data to reproduce our experiments.[1] Our contributions in this work are threefold:

1. We empirically demonstrate *self-recognition*: even with very short texts in relatively low-entropy scenarios, LLMs can reliably distinguish their own output from human-written content.

2. We introduce a steering-based watermarking technique that injects a distinct, recoverable signature, enabling

accurate attribution across multiple identical LLM instances steered in different directions, without compromising generation quality.

3. We offer an analysis that sheds light on the capacity of internal representations to encode and retrieve a random signal.

## 2. Methods

### 2.1. Problem Setup

We consider *white-box* detection and attribution of LLM-generated text using internal activations of a target model $M$. We distinguish two setups:

**(i) AI-generated text self-recognition**: Given a text $t$, decide whether it was generated by $M$ or written by a human.

**(ii) Multi-model attribution**: Decide which *steered* variant of the same base model $M$ produced a given text $t$.

**Threat model**: At detection time, we assume white-box access to the internal activations of $M$ when processing a given text $t$. We consider both a *prompt-conditioned* setting, in which the prompt is available to the detector, and a *prompt-agnostic* setting, in which only the generated text $t$ is observed.

### 2.2. Activation Extraction

Given a model $M$ with $L$ layers, let $l \in \{1, \ldots, L\}$ denote a layer and $\mathbf{x} = [x_0, x_1, \ldots, x_{n-1}]$ denote the token sequence used for activation extraction (prompt + completion, or completion only in the prompt-agnostic setting). Let $\mathbf{A}_l(x_i) \in \mathbb{R}^d$ be the intermediate activation at layer $l$ downstream from token $x_i$. By extension, for a text $t$, we describe the activation sequence as $\mathbf{A}_l(t) \in \mathbb{R}^{n \times d}$. We extract $\mathbf{A}_l(\cdot)$ at a fixed, consistent location within each transformer block. The extraction layer $l$ is optimized independently for each model and corresponds to an intermediate layer near the middle of the network.

### 2.3. Steered LLM Generation

To embed a reliably detectable signature in the generated text, we steer the model during generation. Given a directive vector $\boldsymbol{v} \in \mathbb{R}^d$ and an amplification coefficient $\alpha > 0$, we add the scaled vector to the intermediate activation at layer $l$ at each token generation step:

$$\mathbf{A}_l(x_i) \leftarrow \mathbf{A}_l(x_i) + \alpha \boldsymbol{v}.$$

Intuitively, this intervention nudges the internal representation trajectory in a consistent direction, leading to a text $t_{\boldsymbol{v}}$ influenced by $\boldsymbol{v}$. At detection time, this influence can be retrieved in the activations $\mathbf{A}_l(t_{\boldsymbol{v}})$ of $M$.

[1] https://github.com/Thibaud-Ardoin/ LLM-Self-Recognition.git

For simplicity, steering and activation extraction use the same layer $l$. The steering direction $v$ is chosen randomly to create a unique watermark in the multi-model attribution setting. While prior work has largely focused on dense interventions that modify all hidden dimensions (Li et al., 2023), recent studies suggest that sparse steering vectors induce fewer undesired perturbations and lead to more stable model behavior. Such sparsity can be achieved either by leveraging sparse autoencoders (Bayat et al., 2025) or by directly enforcing sparsity in the activation space (Ardoin et al., 2025). We adopt the latter approach and construct steering vectors with only a small fraction of non-zero dimensions. We empirically evaluate this design choice in Section 3.4.

### 2.4. Attribution

To highlight the intrinsic self-recognition capability of LLMs, we adopt a minimal linear probe operating on averaged internal activations. Multi-model attribution, however, poses a more challenging classification problem and therefore needs a more expressive probing approach.

#### 2.4.1. LINEAR CLASSIFICATION FOR SELF-RECOGNITION

To obtain a fixed-dimensional representation of a text $t$ that is independent of its token length $n$, we aggregate the token-level activations extracted in Section 2.2 by simple averaging:

$$\mathbf{r} = \frac{1}{n} \sum_{i=0}^{n-1} \mathbf{A}_l(x_i) \in \mathbb{R}^d \qquad (1)$$

We use a lightweight linear discriminant analysis (LDA) classifier to distinguish *human* from *LLM-generated* text based on the extracted, standardized representations $\mathbf{r}$. LDA gives an affine decision function, which we use to score texts and classify them using a threshold $\tau$ selected to satisfy a target false-positive rate. In high-dimensional regimes, the empirical covariance estimate of the LDA may be ill-conditioned. We therefore regularize the covariance with the Ledoit–Wolf shrinkage estimator (Ledoit & Wolf, 2004). In this setting, we use an 80/20 train/test split.

#### 2.4.2. MULTI-MODEL ATTRIBUTION

To learn to distinguish among $K$ different steering vectors $(v_k)_{k \in [1,K]}$, we construct a labeled dataset of activations. Let $M_{v_k}(p) = t_{v_k,p}$ denote the text generated by model $M$ steered with vector $v_k$ in response to prompt $p \in \mathcal{P}$:

$$\mathcal{D} = \{(\mathbf{A}_l(t_{v_k,p}), k) \mid k \in \{1, \ldots, K\}, \ p \in \mathcal{P}\}.$$

We divide the dataset $\mathcal{D}$ with a 70/10/20 train/validation/test split by partitioning the prompt set $\mathcal{P}$, ensuring no prompt overlap across splits. We then train a multi-layer perceptron (MLP), which provides greater expressive capacity than LDA. The MLP has two hidden layers of width 32, trained for a single epoch to predict the steering index $k$ of a given extracted activation. As detection operates at the *token level*, it enables attribution on arbitrary-length texts. However, tokens later in a sequence encode broader context due to causal attention over preceding tokens. To obtain *text-level* predictions, we aggregate token-level classifications by majority voting across the entire generated sequence.

## 3. Experiments

Our experiments aim to assess the effectiveness of using internal activation patterns for text authorship classification and to validate the proposed steering-based watermarking method. We primarily conduct our experiments using `Llama-3.1-8B`, chosen for its wide availability and favorable performance-to-size trade-off. To assess the generality of our approach across model families, we additionally evaluate `Ministral-3-8B`, a state-of-the-art mixture-of-experts (MoE) model in the 8B-parameter range. To study scalability, we further include experiments on the smaller `Llama-3.2-1B` and `Llama-3.2-3B` models. For question-answering benchmarks, we use the instruction-tuned variants of all models.

To evaluate robustness across a range of generation regimes, we employ diverse datasets ranging from low-entropy constrained outputs to high-entropy open-ended completions. For low-entropy tasks designed to demonstrate the self-recognition capability of LLMs, we use the XL-Sum summarization dataset (Hasan et al., 2021). We restrict our experiments to the English subset of the human-authored BBC news articles and summaries. For long-form question answering, we use the ELI5 dataset (Guo et al., 2023), consisting of open-ended questions extracted from the *Explain Like I'm Five* subreddit. Finally, to evaluate performance under higher-entropy generation, we curate a *Fresh News* dataset from news articles published after the training cutoff of the evaluated models.

### 3.1. Self-Recognition of Unaltered LLM-Generated Text

To evaluate the capability of LLMs to distinguish texts they generated themselves from human-authored ones, we prompt the LLM to generate short summaries of given news articles and compare them against the corresponding human-authored summaries. This setting enables tight control of confounds, particularly regarding text length. Furthermore, summarizing news articles in only 1–2 sentences is inherently low-entropy, as the summary must cover the article's key information, leaving little stylistic freedom. Authorship attribution in this setting is therefore more challenging

*Table 1.* Self-recognition AUROC scores (%) comparing our activation-based classifier (layer index in parentheses) against a perplexity baseline on the XL-Sum dataset. Both methods achieve near-perfect performance when conditioned on the original prompt, but only our method generalizes to the prompt-agnostic setting. Higher is better.

| | **With prompt** | | **No prompt** | |
| Model | Ours | PPL | Ours | PPL |
|---|---|---|---|---|
| Ministral-3-8B | 100 | 99.71 | 99.99 | 32.33 |
| Llama-3.1-8B | 99.99 | 99.19 | 99.16 | 47.86 |
| Llama-3.2-3B | 99.96 | 99.43 | 99.03 | 47.49 |
| Llama-3.2-1B | 99.82 | 97.07 | 98.58 | 52.27 |

*Table 2.* F1 scores of the attribution task on ELI5 and Fresh News datasets across multiple LLMs. Results are averaged over five runs with different random seeds and steering vectors. The task measures the ability to classify texts generated by two independently steered model variants. Higher is better.

| | **ELI5** | | **Fresh News** | |
| Model | Token | Text | Token | Text |
|---|---|---|---|---|
| Ministral-3-8B | 99.3 | 100 | 94.6 | 100 |
| Llama-3.1-8B | 94.0 | 99.1 | 90.5 | 99.1 |
| Llama-3.2-3B | 83.3 | 95.5 | 75.5 | 88.3 |
| Llama-3.2-1B | 72.0 | 85.3 | 69.5 | 83.8 |

than in conventional free-form text-generation tasks, where AI-GTD is typically evaluated.

In this experiment, we evaluate self-recognition using $8\,192$ news articles and reference summaries from the English subset of the XL-Sum dataset (Hasan et al., 2021). We consider only articles at most $2\,048$ characters long to reduce arbitrariness in the summaries and to allow LLMs to produce high-quality summaries for each article.

As a baseline, we use each LLM's perplexity (Jelinek et al., 2005) as an intrinsic measure of its predictive performance. For a token sequence $x_{1:T}$, we compute

$$\mathrm{PPL}(x) = \exp\left(-\frac{1}{T}\sum_{t=1}^{T}\log p_M(x_t \mid x_{<t})\right),$$

where $p_M$ denotes the conditional probability distribution defined by model $M$. Lower perplexity indicates that the model assigns higher likelihood to the text. We perform authorship attribution by scoring each summary with its perplexity and sweeping a decision threshold over the full range of scores to compute the AUROC, expecting the generating model to assign lower perplexity to its own summaries than to human-authored ones. While perplexity is commonly used in AI-GTD, it is a limited signal and is often augmented with additional statistics or computed in a contrastive fashion. For example, Hans et al. (2024) propose a zero-shot detector that contrasts two closely related LLMs by combining perplexity with a *cross-perplexity* score, which helps calibrate prompt- and topic-induced variation. We nevertheless report perplexity because it is computable from a single LLM without auxiliary models or features, making it a natural baseline for self-recognition.

Table 1 presents self-recognition capabilities across LLMs in this scenario. Our findings indicate that internal activations exhibit a strong, linearly separable signal for distinguishing LLM-generated from human-authored summaries, even for very short completions. Classifiers trained on these activations consistently outperform the perplexity-based

baseline, though the baseline performs only marginally worse when prompts are available. Crucially, the activation-based signal remains robust in the prompt-agnostic setting, with AUROC decreasing by at most one point. By contrast, for Llama-family models, perplexity-based baselines do not consistently perform above chance without prompt access. Notably, for Ministral-3-8B-Instruct in this setting, the perplexity ordering is largely inverted, as human-authored summaries tend to attain *lower* perplexity than the model's own generations. This behavior may reflect data overlap between the evaluation corpus and the pre-training or instruction-tuning data used for Ministral-3-8B-Instruct.

Additional experiments described in Appendix E indicate that layer-wise LDA performance remains largely consistent beyond the first layer, suggesting that self-recognition signals are distributed throughout the network. Further, classifiers trained on specific news domains generalize well to others, indicating that the learned signal is not strongly confounded by semantics.

### 3.2. Distinguishing Multiple LLMs with Steering

To validate the feasibility of the activation-based watermark described in Section 2.4.2, we conduct a controlled attribution experiment. We steer two identical model instances using two distinct random sparse vectors to generate completions for a fixed set of $1\,000$ prompts, with a maximum sequence length of 512 tokens. A classifier is then trained on the collected activation data to distinguish the source identity of the generated text.

To isolate the intrinsic detectability of the steering signal from extensive hyperparameter optimization, we fix the amplification coefficient to $\alpha = 5$ and the vector sparsity to $99.7\%$ across all model families. While these parameters may not be optimal for every specific architecture, fixing them enables a direct comparative assessment of signal recovery across scales. Table 2 presents the attribution accuracy, revealing a clear trend: detection perfor-

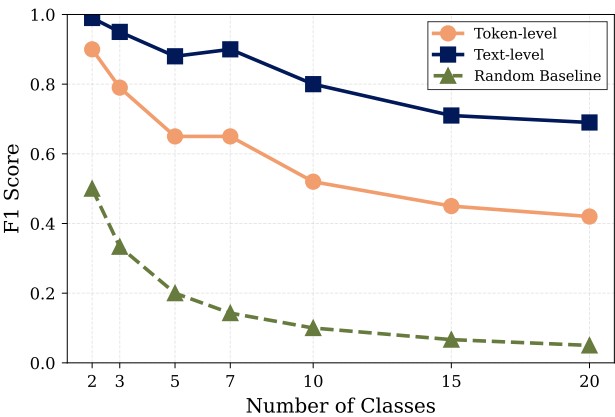

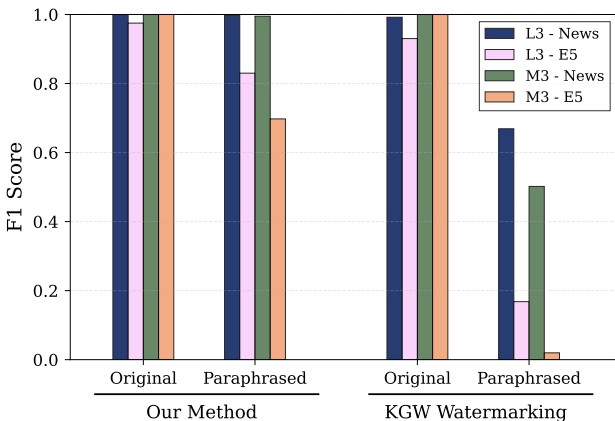

*Figure 2.* Performance of our attribution method across different numbers of classes. We compare text-level attribution with majority voting, token-level attribution, and a random-attribution baseline. F1 scores are shown for `Llama-3.1-8B` on our Fresh News dataset.

*Figure 3.* F1 scores for AI-GTD, comparing our method with traditional watermarking on the original LLM-generated text and a paraphrased version. Scores are computed with `Llama-3.1-8B` (L3) and `Ministral-3-8B` (M3) on the Fresh News (News) and ELI5 (E5) datasets.

mance improves with model size. Interestingly, the question-answering dataset ELI5 yields higher separability than open-ended text generation tasks. This finding is counterintuitive, as the lower entropy of factual generation theoretically offers a more constrained manifold for embedding a watermark. This performance gap could therefore stem from the instruction fine-tuning of the underlying models.

We further evaluate the method's scalability by increasing the pool of distinct steered identities. As shown in Figure 2, while the discrimination task becomes harder as the number of classes grows, performance remains significantly above the random baseline. While the current classification over $N$ vectors shows limitations in scaling, these results imply the potential for a combinatorial multi-bit watermark, where multiple sparse vectors are superimposed for steering and detected independently. This would theoretically allow the encoding of $2^N$ identities.

Regarding generation quality, perplexity in this setting acts primarily as a measure of distributional shift from the base model rather than a direct proxy for text quality. To ensure that steering does not degrade generation quality, we calibrate the steering parameters using an external evaluator. We employ the `quality-classifier-deberta` model, a fine-tuned `DeBERTa-v3` (He et al., 2023) from the NVIDIA NeMo Curator toolkit. In addition, we evaluate the steered models on the MMLU benchmark (Hendrycks et al., 2020) to verify minimal performance degradation. A comprehensive analysis of the quality trade-off is provided in Appendix F.

### 3.3. Robustness Study

AI-GTD often lacks robustness to modifications of the generated text, which can erase the watermark or the

LLM's fingerprint. Automatic paraphrasing tools, such as `DIPPER-XXL` (Krishna et al., 2023), have been designed to evaluate this robustness and are used here. `DIPPER-XXL` reformulates text at the level of multiple sentences, rather than relying only on word substitution or sentence-level rewriting. Figure 3 reports F1 scores for distinguishing human-written from watermarked text, with and without paraphrasing, and compares our method to the green-list approach of the KGW watermark (Kirchenbauer et al., 2023). Our method shows greater robustness to paraphrasing, particularly in the free-form generation setting, where performance remains largely stable. One possible explanation is that our approach operates in a higher-level representation space, where attribution signals are aggregated across many tokens, potentially leading to more robust detection.

Here, we paraphrase only the LLM-generated text to wash out the watermark or steering signal. If we also paraphrase the human text, the F1 score of our method is slightly reduced, as paraphrasing human text acts as a spoofing mechanism for our detector. More details are given in Appendix C.

### 3.4. Influence of Steering-Vector Sparsity

Excessive steering intervention introduces significant noise into the generation process, often manifesting as repetitive loops or incoherent text. While these distinct artifacts make the attribution task trivial, they render the watermark unusable for practical applications where text quality is essential. Consequently, the optimal steering configuration must maximize detectability while minimizing quality loss.

To demonstrate the empirical advantages of sparse steering vectors over dense ones, we evaluate the trade-off they offer between attribution performance and generation qual-

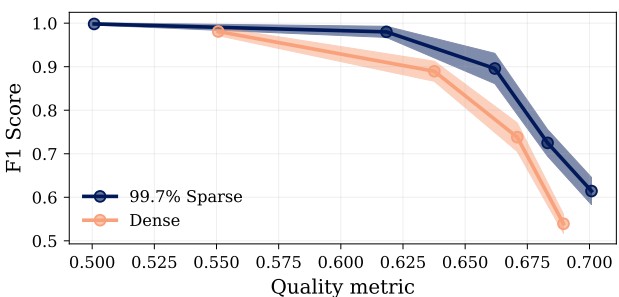

*Figure 4.* Trade-off between generated text quality and accuracy in distinguishing vanilla from steered generations of `Llama-3-8B`. The figure compares the effect of a 99.7% sparse steering vector with a dense counterpart. Points are obtained by varying the steering coefficient and are averaged over five random seeds, with all other settings held fixed.

*Table 3.* Accuracy for source attribution at the token and text levels, with and without paraphrasing. We compare simple cosine similarity attribution with the trained MLP approach under identical evaluation conditions. Higher is better. The best score for each method is underlined.

| Detection Method | Granularity | Original Text | Paraphrased Text |
|---|---|---|---|
| Similarity | Token-level | 62.9 | 58.6 |
| Similarity | Text-level | 84.6 | 77.8 |
| MLP | Token-level | 90.5 | 80.0 |
| MLP | Text-level | 99.1 | 89.3 |

ity. Since the multi-model attribution task described in Table 2 yields saturated accuracy for large models, we adopt a more challenging binary classification setting to better differentiate the steering types: distinguishing the unsteered `Llama-3-8B` base model from a steered counterpart. We perform a sweep over the amplification factor $\alpha$, averaging the performance across five distinct steering vectors per configuration. The quality metric is computed using the `quality-classifier-deberta` model. The results in Figure **??** indicate that highly sparse vectors achieve a favorable trade-off profile compared to their dense counterparts, despite modifying only a fraction of the dimensions. This suggests that broad interventions are unnecessary, highlighting that targeted activation engineering offers a more robust approach to model steering.

### 3.5. Direct Signal Recovery via Cosine Similarity

Unlike standard activation engineering, which identifies specific directions to inhibit or amplify behavioral traits, such as sentiment or hallucination, our method operates on a fundamentally different principle. We inject a random, sparse steering vector. In the high-dimensional geometry of LLM activations, random vectors are, with high probability, approximately orthogonal to the semantic manifolds governing standard model behaviors, a property of concentration of measure (Vershynin, 2026). Therefore, we hypothesize that this orthogonality allows the steering signal to coexist with semantic content without significant interference.

Remarkably, this steering signal persists through the discretization step of token sampling and re-embedding. When the generated text is fed back into a non-steered base model, the steering vector can be directly retrieved from the resulting activations, without requiring a trained classifier. We observe a significant cosine similarity between the original steering vector and the activations of the base model processing the watermarked text. This implies that the trans-

formation from activation space to discrete tokens and back to activation space preserves the directional alignment of the injected signal. As illustrated in Figure 5, this alignment remains robust even after strong paraphrasing via `DIPPER-XXL`. The figure reports the average cosine similarity across all tokens in sequences up to 512 tokens.

This persistence enables a training-free detection mechanism. Instead of training a supervised MLP, we can attribute a text to a specific steered model by computing the cosine similarity between the collected activations and candidate steering vectors. This yields an attribution score for each token and, via majority voting, a global attribution decision. Table 3 presents the accuracy of this zero-shot approach for distinguishing between two random vectors. While it does not reach the near-perfect performance of the trained MLP, the accuracy is surprisingly high for a simple geometric metric. This confirms that the sparse signal is not merely a statistical artifact but is physically encoded into the generated sequence.

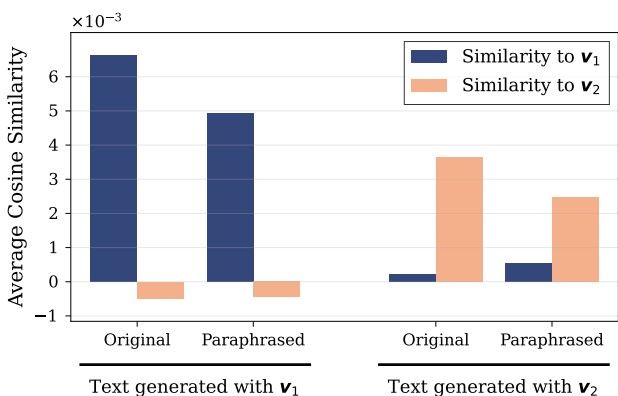

*Figure 5.* Example of alignment between steering directions and representations of induced text. Averaged cosine similarity between a pair of randomly sampled steering vectors ($v_1$ and $v_2$) and activation representations extracted from text generated by the model when steered along these directions.

# 4. Discussion and Future Work

**Choice of Steering Vectors.** An analysis of individual steering directions reveals substantial variability in their effects. Some vectors induce more severe degradations in generation quality, while others are more easily detectable. Steering directions also differ in their impact on the representation space at detection time, as measured by cosine similarity in Figure 5. We hypothesize that these differences arise from varying degrees of orthogonality to the model's natural activation trajectory in a given context. This heterogeneity suggests that selecting or optimizing subsets of steering vectors that balance detectability and generation quality could further improve performance.

**Multi-bit Watermarking.** As proposed in Section 3.2, superimposing multiple sparse vectors with disjoint supports theoretically enables a high-capacity, multi-bit watermark. In such a scheme, the presence or absence of a specific steering vector encodes a single bit of the unique identifier. Achieving this requires further research to verify that the linear superposition of steering signals remains recoverable within the model's activation space without destructive interference.

**Interpretability of Random Steering.** To the best of our knowledge, utilizing sparse random vectors is a novel approach in activation engineering, specifically for watermarking. While our metrics indicate negligible degradation in quality, the semantic nuances induced by these perturbations remain unclear. We could not identify obvious artifacts in the text output, yet a subtle shift in distribution is inevitable to allow detection. Future work must rigorously analyze these impacts, potentially leveraging automated annotation by strong judge models or interpreting the steered features via sparse autoencoders for monosemantic feature discovery (Templeton et al.). Such interpretability is a prerequisite for deployment, ensuring that the watermark does not inadvertently introduce safety risks or biases while helping to design more semantically neutral steering vectors.

Cross-evaluation results displayed in Appendix G reveal a hard architectural boundary: detection remains strong within the same family but degrades to chance level when text is generated by another architecture, regardless of dataset. This mirrors observations of the *subliminal learning* phenomenon reported by Cloud et al. (2025), where implicit behavioral traits in teacher-generated data transfer to a student only when both models share the same base architecture. We interpret both findings as evidence of architecture-specific activation structure: steering vectors leave activation traces that are invisible at the semantic level but geometrically coherent within a model family, and do not generalize beyond it.

**Robustness and Secrecy.** While we demonstrated robustness against strong paraphrasing, the watermark's resilience to broader attacks, such as recursive translation, adversarial perturbations, text degradation, or format changes, remains to be characterized. Furthermore, the security of the method relies on the secrecy of the steering configuration. Specifically, if an attacker can recover the secret steering vector and target layer, the watermark could be trivially reproduce to spoof the identity of the steered LLM. To mitigate this risk, future work could explore dynamic steering strategies in which the vector rotates or evolves during generation according to a pseudo-random schedule. Such a moving target would significantly complicate the recovery of the watermark signal without the corresponding seed.

# 5. Limitations

**Detection Overhead.** Unlike inference-time watermarks, signal injection incurs no computational cost in our method. However, detection requires white-box access to the model and a forward pass to extract activations from the text, which incurs a non-negligible computational cost. In our setup using an RTX A5500, collecting the model's activations takes time equivalent to $21\%$ of the time needed to autoregressively generate the same text.

**Quality Evaluation Limitations.** To obtain a more accurate evaluation of textual quality than perplexity, we relied on an externally fine-tuned quality assessment model. However, this evaluator exhibits biases toward structural formatting, such as penalizing markdown syntax regardless of content quality. Theoretically, degraded texts may still be assigned high scores, potentially obscuring small performance losses in steered outputs. Despite these limitations, we maintain that measuring the *relative* quality difference between steered and unsteered outputs provides a valid proxy for steering-induced degradation, as formatting and other confounders remain constant across conditions.

# 6. Related Work

Recent work on mechanistic interpretability shows that transformer activations encode linearly recoverable signals for high-level concepts. In the context of truthfulness, simple probes of hidden states can predict the veracity of a statement even when the model's final answer is incorrect, suggesting concept directions that are partially disentangled from generation (Marks & Tegmark, 2023; Azaria & Mitchell, 2023).

Several approaches use internal representations for detecting AI-generated text. RepreGuard (Chen et al., 2025) separates human and model text using observer-model hidden states, while Yu et al. (2024) train supervised classifiers on acti-

vations from an auxiliary encoder. These works primarily target robust, prompt-agnostic detection of arbitrary LLM-generated text, rather than self-recognition. SAEMark (Yu et al., 2025) uses SAE features to guide a rejection-sampling process for watermarking, achieving robust detection at the cost of significant computational overhead during generation.

Closest to our setting, Ackerman & Panickssery (2025) study whether instruction-tuned models can recognize their own outputs and extract residual-stream directions via contrastive pairs, but report performance that does not consistently beat perplexity-based heuristics. Separately, Kuznetsov et al. (2025) show that SAE-based features can improve general-purpose AI-GTD. They use feature steering to analyze the nature of the features responsible for detection and causally link the intervention to detectability.

## Conclusion

In this work, we investigated the self-recognition capabilities of LLMs and introduced a random steering mechanism to enable the attribution of generated text. Our results show that LLMs exhibit strong self-recognition capabilities, encoded as linearly separable signals in their internal activations. These signals are distributed across the network, robust to semantic shifts, and enable attribution without relying on perplexity.

Furthermore, a simple inference-time intervention enables robust source attribution while incurring minimal degradation in text quality. Importantly, we find that sparse interventions in the activation space lead to lower quality loss than dense noise injection for comparable detectability. Moreover, by operating directly on model activations, our approach avoids reliance on computationally intensive sparse autoencoder pipelines while still capturing high-level properties of the inference process.

Future work may extend this analysis to the design of a seamless, resource-efficient watermarking system, as well as to systematic comparisons with existing approaches. More broadly, we hope this work encourages further investigation into the inference-time dynamics of LLMs and into how internal representations can be leveraged to ensure reliable and transparent model behavior.

## Impact Statement

This paper examines whether LLMs encode a reliable self-recognition signal in their internal activations and leverages it to enable watermarking and attribution at the level of a model or a steered variant. Such capabilities could support provenance, auditing, and accountability for AI-generated text, but they also amplify known risks. In particular, water-

marking for attribution can raise privacy concerns if signatures directly link to specific deployments or users.

We identify no substantive risks associated with the publication of our method, and our contribution is purely methodological. The threat models we evaluate are standard in the watermarking literature and can be executed using publicly available tools. Consequently, disclosing our method does not introduce any new adversarial capabilities beyond those already well known in existing watermarking frameworks.

## Author Contributions

TA and JS contributed both significantly to this work. TA led the steering experiments and writing, JS led the self-recognition experiments, under the general direction of GW.

## Acknowledgements

TA, JS and GW were supported by the Federal Ministry of Education and Research of Germany (BMBF) in the programme of "Souverän. Digital. Vernetzt.", joint project "AIgenCY : Chances und Risks of Generative AI in Cybersecurity", project identification number 16KIS2013. GW was also supported by BMBF joint project "6G- RIC: 6G Research and Innovation Cluster", project identification number 16KISK025.

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

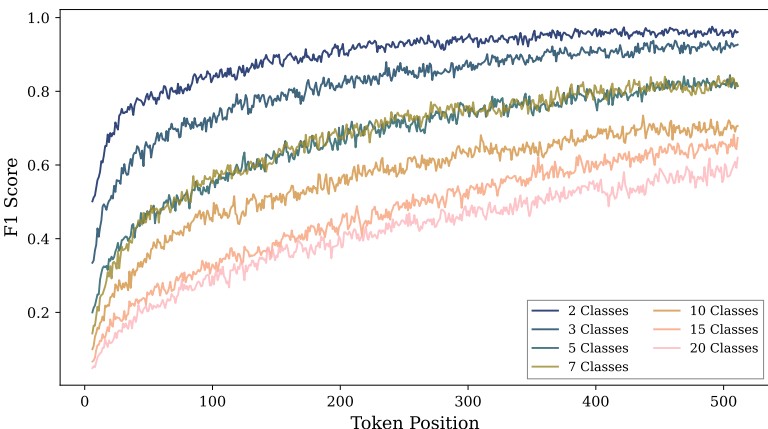

*Figure 6.* Test F1 score as a function of token position, with attribution tasks involving between 2 and 20 distinctly steered LLMs.

## A. Parameterization

This section contains the parameters used to enable reproduction of our results. More details can be found in the code repository (https://github.com/Thibaud-Ardoin/LLM-Self-Recognition).

**Datasets.** The shared repository also contains the text datasets used in our experiments. The custom-curated **Fresh News** dataset was collected via the API of *The Guardian* and consists of clean, plain-text news articles. To minimize potential overlap with model training-data cutoff dates, all articles were published between November 1, 2025 and January 5, 2026. As the dataset is designed for open-ended text completion, we retain only the first six words of each article as the initial prompt provided to the LLM. The **ELI5** dataset was filtered to remove questions containing explicit mention of the word "Edit", which typically correspond to user-added clarifications appended after the initial question. Questions containing explicit "URL" mentions were also removed, as they often refer to external content that is not accessible to the LLM. For the robustness experiments in Section 3.3, the human reference answer is consistently chosen as the first response listed in the dataset.

**Models.** Texts are generated with nucleus sampling ($p = 0.9$) and temperature $0.7$ for all LLMs and datasets, using the system prompt and query template specified in Appendix D. In the steering scenarios, a repetition penalty is set to $1.1$.

**Steering.** The random sparse steering vectors are sampled from $\mathcal{U}([-1, 1]^d)$, with $99.7\%$ of dimensions set to zero. The scaling coefficient $\alpha = 5$ and the sparsity ratio are selected based on preliminary experiments to balance detection performance and generation quality on `Llama-3.1-8B-Instruct`. Injection and detection are performed at the middle layer, which empirically yields robust performance across models.

## B. Token-wise Analysis

Figure 6 illustrates how the test-time performance varies as a function of token position in the generated sequence. The experimental setup is the MLP classification described in Section 2.3. The earliest tokens are the most difficult to classify, with accuracy close to random chance $\frac{1}{N}$ when distinguishing among $N$ steered LLMs. In contrast, later tokens achieve substantially higher accuracy, approaching the performance obtained by majority voting. While performance could be improved by excluding early tokens from the majority vote, this would break the sequence-length-agnostic nature of the method.

Figure 7 illustrates the cosine similarity between generated text representations and the steering vectors used during generation. With the vector active during steering, we observe an increasing trend similar to that seen in the test accuracy of the MLP setup. For paraphrased texts, the magnitude of this effect is reduced, while the overall trend remains consistent. In contrast, cosine similarity with steering vectors that were not involved in generating the text oscillates around zero or takes negative values.

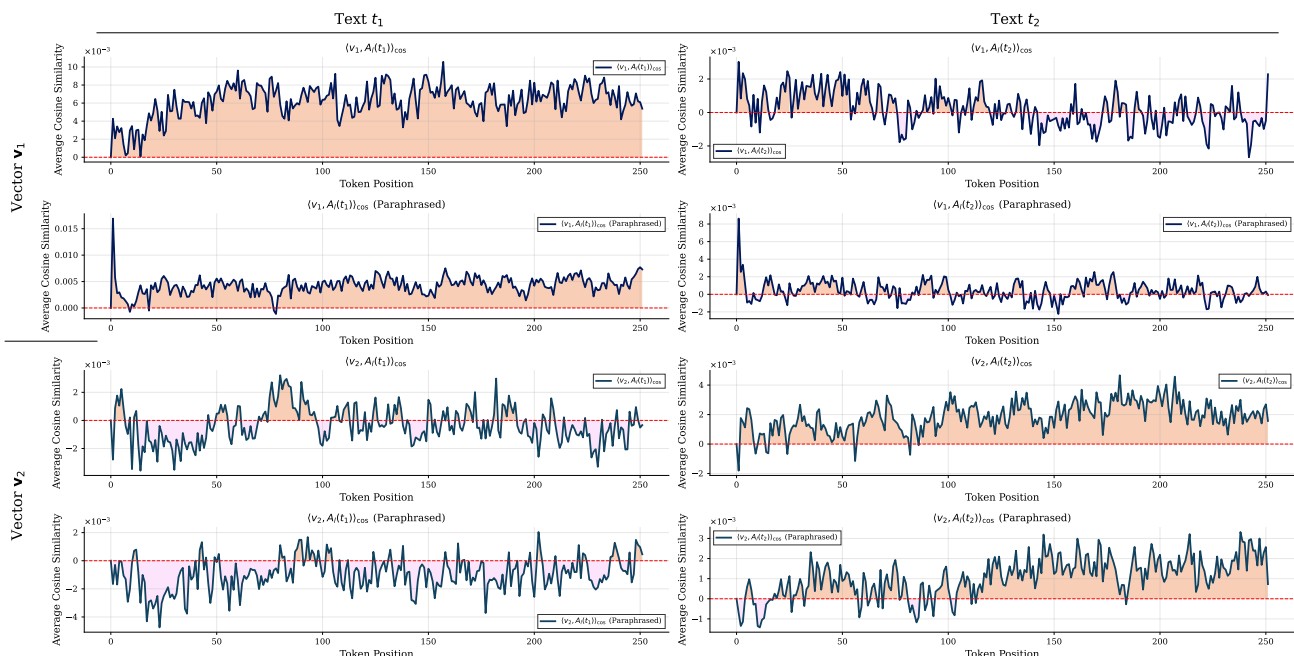

*Figure 7.* Variation in cosine similarity $\langle \cdot, \cdot \rangle_{\cos}$ as a function of token position. The plots compare the similarity between activations of texts $t_1$ and $t_2$, generated with steering vectors $\mathbf{v}_1$ and $\mathbf{v}_2$, respectively. The paraphrased versions of the texts are also compared.

## C. Details of the Paraphrasing Experiments

The paraphrasing model `DIPPER-XXL` (Krishna et al., 2023) is used with the parameters *Lexical diversity* $= 60$ and *Reordering* $= 20$. These parameters are among the strongest setups proposed in the original paper.

Figure 8 presents the results of an alternative robustness experiment to the one in Section 3.3. Because paraphrasing only the watermarked text introduces a distribution shift, we additionally evaluate a setting in which both human- and AI-generated texts are paraphrased. While the KGW watermark is largely unaffected under this new setup, our method exhibits a reduction in accuracy relative to the results shown in Figure 3. This suggests that AI-generated paraphrasing acts as a spoofing mechanism for our representation-based attribution methods.

## D. Prompts

Below are the system and user prompts used to evaluate the general self-recognition ability of LLMs (Section 3.1). The prompts were designed to generate summaries that closely match the human-authored ones.

### D.1. System Prompt

```
You write very short summaries of news articles.
Rules:
- Output only one sentence whenever possible (but never more than two).
- Respect the requested word range.
- The final sentence must end with a single period.
- No preface, labels, quotes, bullet points, or line breaks.
- Use a neutral, journalistic tone.
- Focus on the main event and key entities only.
```

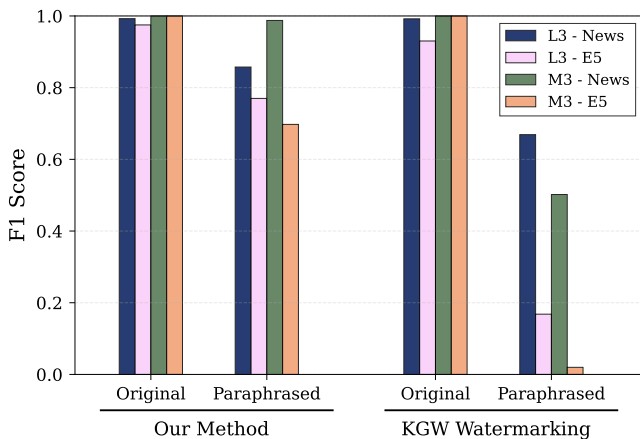

*Figure 8.* Accuracy for AI-GTD, comparing our method with traditional watermarking on the original LLM-generated text and a paraphrased version. Accuracies are computed with `Llama-3.1-8B` (L3) and `Ministral-3-8B` (M3) on the Fresh News (News) and ELI5 (E5) datasets.

*Table 4.* Layer-wise self-recognition performance summary. For each model and setting, we report the best- and worst-performing layer indices, along with the AUROC spread (difference in percentage points).

| Model | With prompt | | | No prompt | | |
|---|---|---|---|---|---|---|
| | **Best L** | **Worst L** | **ΔAUROC** | **Best L** | **Worst L** | **ΔAUROC** |
| `Ministral-3-8B` | 13* | 0 | 0.07 | 13 | 29 | 0.07 |
| `Llama-3.1-8B` | 12 | 0 | 2.39 | 15 | 0 | 1.81 |
| `Llama-3.2-3B` | 10 | 0 | 2.39 | 14 | 0 | 1.59 |
| `Llama-3.2-1B` | 6 | 0 | 1.44 | 8 | 0 | 1.42 |

*Note:* * indicates the smallest layer index was chosen in the case of ties.

## D.2. Query

```
Summarize the article in one sentence or two at most (10–25 words).
Prefer a single sentence if possible.
No preface, labels, quotes, bullet points, or line breaks.

ARTICLE START
{text}
ARTICLE END

Answer:
```

## E. Additional Experiments Assessing Self-Recognition

**Localization of Self-Recognition Signals.** Across all four LLMs, layer-wise LDA performance is highly consistent (Table 4). The performance spreads reported in the table are computed over all layers and are largely driven by layer 0, which is typically the weakest. Excluding layer 0, performance is nearly flat: the within-model spread is at most $0.56$ AUROC points with prompts and at most $1.2$ points without prompts. The best-performing layers generally occur near the middle of the network. `Ministral-3-8B-Instruct` exhibits negligible spread in both settings, plausibly because its AUROC is already near the ceiling on this benchmark (Table 1). For reference, `Llama-3.2-1B` has 16 layers, `Llama-3.2-3B` has 28, `Llama-3.1-8B` has 32, and `Ministral-3-8B` has 34.

**Out-of-Distribution Performance.** Table 5 reports classification AUROC for probes evaluated on internal activations from `Llama-3.1-8B-Instruct`, extracted from summaries drawn from different news domains. For each source domain, we train the classifier on 80% of the available in-domain samples and reserve the remaining 20% for in-domain evaluation.

*Table 5.* Out-of-distribution AUROC scores (%) for texts and activations from `Llama-3.1-8B-Instruct`. Each row is the source domain used to train the LDA classifier; each column is the target domain used for evaluation. In-domain results are italicized. Higher is better.

| | Evaluation domain | | | | | |
|---|---|---|---|---|---|---|
| **Train domain** | **All** | **World** | **UK** | **Tech** | **Busin** | **Enter** |
| **All** | *99.99* | 99.93 | 99.99 | 99.95 | 99.94 | 99.87 |
| **World** | 99.98 | *100.00* | 99.98 | 99.98 | 99.95 | 99.92 |
| **UK** | 99.99 | 99.90 | *100.00* | 99.93 | 99.92 | 99.78 |
| **Technology** | 99.74 | 99.84 | 99.72 | *100.00* | 99.88 | 99.49 |
| **Business** | 99.90 | 99.88 | 99.92 | 99.93 | *100.00* | 99.72 |
| **Entertainment** | 99.87 | 99.80 | 99.87 | 99.92 | 99.93 | *99.99* |
| *Support* | 8192 | 8192 | 8192 | 2544 | 6710 | 5569 |

We then evaluate the resulting classifier on all samples from each out-of-domain target domain. As in Section 3.1, we only consider articles that are at most $2\,048$ characters in length, although this results in smaller support for less represented domains in XL-Sum. Overall, performance remains high under domain shift, indicating that the learned signal generalizes across topics and is not primarily driven by domain-specific semantic content.

**Completion Length Analysis.** In the experiment described in Section 3.1, human-authored summaries average 25.7 tokens (SD 8.85). Among the models, `Llama-3.2-1B-Instruct` matches this length most closely (mean 27.1, SD 6.23), whereas `Llama-3.2-3B-Instruct` (mean 30.8, SD 4.20) and `Llama-3.1-8B-Instruct` (mean 31.0, SD 4.36) produce slightly longer summaries with lower variance. `Ministral-3-8B` generates the longest summaries (mean 44.0, SD 9.97). Although length information is likely reflected in the activations, performance remains high even when summary lengths overlap substantially, suggesting that the classifier primarily relies on stylistic cues beyond length.

**Stricter Confound Controls.** We further evaluate whether the self-recognition classifiers in Section 3.1 rely on simple formatting or length cues. For this control, all summaries are converted to lowercase and stripped of punctuation. We additionally retain only paired human-written and LLM-generated summaries whose character lengths fall within the closest $10\%$ of pairs, corresponding to an absolute length difference below 22 characters. This filtering is applied separately for each XL-Sum subset.

*Table 6.* Change in self-recognition classification accuracy under stricter confound controls. Each entry reports the difference, in percentage points, relative to the corresponding evaluation without the additional lowercase, punctuation-removal, and length-matching controls.

| Subset | All | Business | Entertainment | Technology | UK | World |
|---|---|---|---|---|---|---|
| $\Delta$ Accuracy (pp) | $-0.20$ | $+0.31$ | $-0.41$ | $-0.22$ | $-0.24$ | $-0.47$ |

As shown in Table 6, the additional controls have only a small effect on classification accuracy across domains. The largest observed change is below $0.5$ percentage points, and performance remains high in the subset of examples where human-written and LLM-generated summary lengths overlap closely. These results suggest that the activation-based self-recognition signal is not primarily explained by casing, punctuation, or summary length. This is consistent with the experimental setting, where summaries are typically only one or two sentences long and generated under low- to moderate-entropy constraints.

## F. Quality Evaluation Metrics

While perplexity is a standard metric in language modeling, it measures the likelihood of a sequence under a specific distribution rather than its semantic quality or utility. In many contexts, PPL is an insufficient proxy for generation quality, for instance, a highly repetitive, degenerate loop often yields low PPL, whereas a creative or novel insight may yield high PPL due to its statistical rarity. To obtain a more robust assessment of text utility, we employ the `quality-classifier-deberta` model from the NVIDIA NeMo Curator toolkit (`https://github.com/NVIDIA/NeMo-Curator`), available through the Hugging Face library (`https://huggingface.co/nvidia/quality-classifier-deberta`). This model is explicitly

fine-tuned to categorize text into "High", "Medium", and "Low" quality tiers. Instead of using the discrete class labels, we extract the softmax probability distribution from the classifier's output to derive a continuous quality score in the range $[0, 1]$, enabling a granular comparison of steering effects. We define the continuous quality score $\hat{q}$ as:

$$\hat{q} = 0 \cdot p_{\text{Low}} + 0.5 \cdot p_{\text{Medium}} + 1 \cdot p_{\text{High}}$$

Using this continuous quality metric, we calibrated the steering parameters to maximize signal strength while maintaining generation utility. The quantitative comparison between the unsteered base models and their steered counterparts is presented in Table 7. In the final row of the table, the relative performance difference indicates that steering induces negligible degradation across tasks. Surprisingly, on the Fresh News completion task, we observe a slight quality improvement of $3.82\%$ in the steered models. Regarding perplexity, while the ELI5 dataset shows an average increase, this is primarily driven by an outlier spike in the steered `Ministral-3-8B-Instruct` model. Manual inspection confirms that this elevated PPL does not correspond to semantic disruption in the generated output, further validating the limitations of PPL as a standalone quality metric.

*Table 7.* Comparison of perplexity and quality metrics between base models and steered versions across two datasets. Values are averaged over 512-token generations for 100 texts and 5 random steering vectors with a sparsity of 99.7% and coefficient $\alpha = 5$.

| | | Fresh News | | ELI5 | |
|---|---|---|---|---|---|
| **Model** | **Version** | PPL $\downarrow$ | Quality $\uparrow$ | PPL $\downarrow$ | Quality $\uparrow$ |
| `Ministral-3-8B` | Vanilla | 5.80 | 0.69 | 8.12 | 0.55 |
| | Steered | 6.39 | 0.71 | 17.99 | 0.48 |
| `Llama-3.1-8B` | Vanilla | 7.62 | 0.69 | 6.92 | 0.65 |
| | Steered | 8.06 | 0.70 | 7.26 | 0.75 |
| `Llama-3.2-3B` | Vanilla | 7.50 | 0.70 | 5.79 | 0.81 |
| | Steered | 7.30 | 0.72 | 5.79 | 0.77 |
| `Llama-3.2-1B` | Vanilla | 9.76 | 0.69 | 8.49 | 0.81 |
| | Steered | 8.27 | 0.74 | 6.85 | 0.80 |
| Averaged diff. (in %) | | -0.52 | 3.82 | 26.72 | -0.90 |

To assess the impact of steering on general LLM performance, we evaluated all models on the full MMLU benchmark (Hendrycks et al., 2020) using the Language Model Evaluation Harness (Gao et al., 2024), comprising $56\,168$ samples. In Table 8, we compare each unsteered (vanilla) model against its steered counterpart, using an identical, non-optimized steering vector across all models. Results are reported under both 1-shot and 5-shot prompting settings.

*Table 8.* MMLU accuracy scores for vanilla and steered models under 1-shot and 5-shot evaluation.

| **Setting** | **Model** | **Vanilla** | **Steered** | Relative $\Delta$ |
|---|---|---|---|---|
| *1-shot* | `Llama-3.1-8B-Instruct` | 65.88% | 65.12% | $-1.16\%$ |
| | `Llama-3.2-3B-Instruct` | 58.46% | 57.81% | $-1.10\%$ |
| | `Llama-3.2-1B-Instruct` | 44.23% | 43.32% | $-2.00\%$ |
| | `Ministral-3-8B-Instruct` | 74.19% | 70.07% | $-5.55\%$ |
| *5-shot* | `Llama-3.1-8B-Instruct` | 67.04% | 66.31% | $-1.10\%$ |
| | `Llama-3.2-3B-Instruct` | 58.83% | 58.40% | $-0.70\%$ |
| | `Llama-3.2-1B-Instruct` | 45.35% | 45.36% | $+0.02\%$ |
| | `Ministral-3-8B-Instruct` | 75.06% | 70.54% | $-6.02\%$ |

The observed performance degradations are consistent but small across the Llama model family, not exceeding $2\%$ in any setting. `Ministral-3-8B-Instruct` shows a comparatively larger relative drop of approximately $5.5\%$ to $6.0\%$. Notably, the steering vector used here is identical for all models and has not been specifically optimized per model. Targeted

optimization could plausibly reduce these degradations. Overall, these results are comparable to quality losses reported for traditional watermarking approaches in previous work (Yang et al., 2025).

## G. Cross-Dataset and Cross-Model Generalization

To assess out-of-distribution robustness, we ran cross-evaluation experiments along two axes. The training pipeline is defined as:

$$\text{train: } Prompt_1 \in \text{D}_1 \rightarrow \text{LLM}_a \rightarrow \text{Text}_{a1} \rightarrow \text{LLM}_a \rightarrow A_{a1} \rightarrow \text{MLP}_a$$

Domain adaptation tests use a different dataset with the same model:

$$\text{test: } Prompt_2 \in \text{D}_2 \rightarrow \text{LLM}_a \rightarrow \text{Text}_{a2} \rightarrow \text{LLM}_a \rightarrow A_{a2} \rightarrow \text{MLP}_a$$

Model adaptation tests use a different generating LLM, with activations extracted from the original training model:

$$\text{test: } Prompt_1 \in \text{D}_1 \rightarrow \text{LLM}_b \rightarrow \text{Text}_{b1} \rightarrow \text{LLM}_a \rightarrow A_{b \rightarrow a} \rightarrow \text{MLP}_a$$

In both cases, activation collection and MLP inference always use the training setup, reflecting a realistic deployment scenario where the provider controls the detection pipeline. We report in Table 9 token-level F1 and aggregated text-level F1 scores for binary classification between two distinct steering vectors.

*Table 9.* Cross-evaluation F1 scores (token-level / text-level) for binary steering vector classification across training and test configurations. Bold diagonal entries indicate in-distribution evaluation.

| Train Setup | | Test Setup | | | | | |
|---|---|---|---|---|---|---|---|
| | | Llama-3.1-8B-Instruct | | Llama-3.1-8B | | Ministral-3-8B-Base | |
| | | **ELI5** | **News** | **News** | **ELI5** | **News** | **ELI5** |
| Llama-3.1-8B-Instruct | **ELI5** | **0.90/0.99** | 0.77/0.84 | 0.74/0.80 | 0.74/0.85 | 0.55/0.52 | 0.58/0.51 |
| Llama-3.1-8B | **News** | 0.76/0.84 | 0.86/0.94 | **0.90/0.99** | 0.78/0.90 | 0.56/0.61 | 0.61/0.65 |

As expected, out-of-distribution performance degrades, though models trained on higher-entropy data generalize better across both domain and architecture shifts, likely due to richer activation variability during training. Notably, the steering signal partially survives even across the Llama-3.1-8B Base and Instruct versions ($\text{LLM}_b \rightarrow \text{LLM}_a$), suggesting that steering leaves traces beyond a single checkpoint within the same architecture. This no longer holds when generating data with a different architecture, such as a Ministral-3-8B model. This matches the results obtained in subliminal learning (Cloud et al., 2025), where subliminal information is transmitted only within similar architectures. Training on a diverse mixture of prompt types and model variants is a natural path to a more robust detector.

**Cross-model transfer for self-recognition.** We similarly evaluate cross-model generalization for the self-recognition classifiers introduced in Section 3.1. For each training model, we extract residual-stream activations from that same model and train a classifier on texts generated by it. At test time, the training model and classifier are kept fixed, but the evaluated texts are generated by a different model. Thus, only the text-generating model changes across columns in Table 10, while activation extraction and classifier inference remain fixed.

We observe that cross-model performance decreases relative to the in-distribution setting, with the strongest degradation appearing when the classifier trained on Ministral-3-8B activations is evaluated on Llama-family generations. In contrast, classifiers trained on Llama-family activations transfer more consistently across Llama-generated texts, suggesting stronger alignment of activation-space signatures within related architectures. These results indicate that self-recognition through residual-stream activations captures model-specific traces that are only partially shared across LLMs. Consequently, cross-model robustness remains limited in this setting, and training on activations induced by a mixture of model families is a natural extension for improving transfer.

*Table 10.* Cross-model classification accuracy for self-recognition based on residual-stream activations. Rows correspond to the model used for activation extraction and classifier training, while columns correspond to the model that generated the evaluated texts. Italicized diagonal entries indicate in-distribution evaluation.

| Train Setup | Test Setup | | | |
|---|---|---|---|---|
| | Ministral-3-8B | Llama-3.1-8B | Llama-3.2-3B | Llama-3.2-1B |
| Ministral-3-8B | *0.9986* | 0.8089 | 0.8068 | 0.7709 |
| Llama-3.1-8B | 0.9865 | *0.9800* | 0.9633 | 0.9354 |
| Llama-3.2-3B | 0.9860 | 0.9645 | *0.9798* | 0.9428 |
| Llama-3.2-1B | 0.9807 | 0.9493 | 0.9510 | *0.9772* |

