# OpenReview forum: "LLM Self-Recognition: Steering and Retrieving Activation Signatures"
_ICML.cc/2026/Conference — ICML 2026 regular_

### Official Review · Reviewer_TZxN · 2026-02-13

**Soundness:** 2
**Presentation:** 3
**Significance:** 3
**Originality:** 3
**Overall Recommendation:** 4
**Confidence:** 4

**Summary:**

This paper studies the question of to what extent a model's activations can encode information about provenance, i.e., whether the model generated the text. The paper studies two settings:

1. Self-recognition, where they show that probes trained on model activations can distinguish between LLM-generated text and human-generated text.
2. More interestingly, the paper studies using activation steering as a form of watermarking, where a sparse steering vector is injected into the model and the model is sampled from as usual. They show that this creates a watermark that can distinguish between multiple variations of the same model with different steering vectors.

They additionally study how well this watermark scales to different numbers of classes, robustness against paraphrasing, the importance of using a sparse steering vector, and the ability to directly recover the original steering vector from the model's activations on the generated text rather than relying on a more involved probing process.

**Compliance With Llm Reviewing Policy:**

Affirmed.

**Final Justification:**

My initial review had 2 primary concerns - 1) limited novelty of the self-recognition results, and 2) limited evaluation of the watermarking results. The author's rebuttals clarified 1), in particular that the results of Ackerman & Panickssery (2025) were limited to instruction tuned models. 2) was resolved with addition experimental results presented both in this rebuttal and the other rebuttals. As such I have increased my rating of significance from 1-> 3, and overall recomendation from 3->4.

**Key Questions For Authors:**

- In light of prior work on self-recognition (Ackerman & Panickssery 2025), can you articulate what novel takeaways your self-recognition results offer beyond what has already been established?
- I find the steering-based watermarking idea genuinely novel and promising. My main hesitation is the limited scope of evaluation. If the authors could provide results on a broader set of benchmarks with more precise quantification of model degradation and comparison to additional baselines, I would be happy to revisit my evaluation.
- Out of curiosity, how well do the probes generalize across tasks? I appreciate the sub-category generalization in Appendix E, but would be curious to see results across different settings, e.g. E5 to news.

**Limitations:**

yes

**Strengths And Weaknesses:**

Strengths:
- The idea of using sparse activation steering as a watermarking mechanism is novel and creative, operating in a fundamentally different space than existing token-level watermarking methods.
- The results showing that a sparse and random vector is best for this procedure, and that the this vector can be directly recovered from the watermarked text are both scientifically interesting.
- The paper is well-written, with clear motivation, and honest limitations.

Weaknesses:
- The experimental evaluation throughout the paper feels limited in scope, with most individual experiments conducted on one or two settings and compared against minimal baselines (perplexity for self-recognition, KGW for watermarking). This makes it difficult to draw strong conclusions about either contribution.
- In light of prior work (Ackerman & Panickssery 2025), the self-recognition results have limited novelty as an existence proof that model activations encode information about self-generated text. The paper suggests this approach naturally extends to distinguishing between different models, but does not demonstrate this. Additionally, W1 make it difficult to assess whether the method has merits purely based on performance as competitive technique for AI-generated text detection.
- The authors acknowledge limitations of their quality metrics, but if the paper's main contribution is a novel watermarking technique, careful evaluation of model degradation is essential. Additionally presenting results relating to model performance on a standard capability benchmarks would significantly clarify the method's impact on generation quality.
- Several of the watermarking results, such as robustness to paraphrasing (Figure 3) and scaling to multiple classes (Figure 2), present detectability as a single-dimensional metric. In practice, any watermarking method involves a tradeoff between signal strength and model degradation, and evaluating these results jointly would be more informative.

---

> ### Author Rebuttal · Authors · 2026-03-31
>
> We thank the reviewer for the enthusiasm toward our approach and for the constructive feedback.
>
> - **Q1: Novelty relative to Ackerman & Panickssery (2025).**
>
> Although our work shares its goal of assessing the capability of self-recognition in LLMs with A&P, it demonstrates that this capability is not limited to instruction-tuned models and even emerges without explicitly querying the LLM about it’s assessed authorship of the text. We therefore challenge the conclusion by A&P that self-recognition capabilities are acquired during post-training, and rather observe them as ubiquitous signals in the LLM’s residual stream. Finally, while A&P fail to consistently outperform perplexity for this task, our method does so, particularly in the prompt-agnostic setting in which perplexity fails to beat chance.
>
> - **Q2: Degradation evaluation**.
>
> To improve the quality evaluation, as you suggested, we ran a full MMLU evaluation (57 subjects, 56,168 samples) comparing vanilla and steered models.
>
>
> | MMLU score accuracy | Llama3.1-8B-Instruct | Llama3.1-8B-Instruct steered | $\Delta$ Llama3.1-8B-Instruct | Llama3.2-3B-Instruct | Llama3.2-3B-Instruct steered | $\Delta$ Llama3.2-3B-Instruct  | Llama3.2-1B-Instruct | Llama3.2-1B-Instruct steered | $\Delta$ Llama3.2-1B-Instruct |
> | --- | --- | --- | --- | --- | --- | --- | --- | --- | --- |
> | 1-shot | 65.88% | 65.12% | -1.16% | 58.46% | 57.81% | -1.1% | 44.23% | 43.32% | -2.0% |
> | 5-shot | 67.04% | 66.31% | -1.1% | 58.83% | 58.40% | -0.7% | 45.35% | 45.36% | -0.02% |
>
> This test revealed that the capability degradation is consistent across model sizes yet small. It is coherent with the degradation reported for other watermarks in the SoK benchmark (Dang et al., 2025). This table will be added along the quality discussion in the paper.
>
> - **Q2: More Baseline comparaison.**
>
> Comparing our method against semantic watermarkings such as SemStamp and multi-bit activation-based SAEMark would indeed be informative. However, both methods proved infeasible to evaluate fairly in our setup: SemStamp's resampling-based generation runs approximately 100× slower than standard inference even using its authors' default parameters and a 7× smaller model, making benchmark-scale evaluation impractical on our hardware. SAEMark's reference implementation contains a infinite resampling loop that prevented any reproducible run.
>
> We were nevertheless able to compare against SynthID (Dathathri et al.), a widely-used token-level watermark. Results on Llama3.1-8B, Fresh News (high-entropy completion), with DIPPER-XL paraphrasing (l=60, o=20):
>
>
> |  | F1 score, no paraphrasing | F1 score Paraphrasing |
> | --- | --- | --- |
> | SynthID (k=5, non-distortionary) | 1.0 | 0.06 |
> | SynthID (k=3, non-distortionary) | 1.0 | 0.23 |
> | KGW (context_width=2) | 0.99 | 0.67 |
> | Steering (alpha=5) | 1.0 | 0.99 |
>
> SynthID collapses under aggressive paraphrasing as a direct consequence of its non-semantic, token-level design. Our activation-space signal is qualitatively more robust, maintaining high accuracy under the same conditions. We will extend this comparison to additional models and datasets in the final version.
>
> Our method is not intended as a drop-in replacement for deployed watermarking systems, but as a proof-of-concept demonstrating that activation space is a viable and robust channel for encoding recoverable, quasi-orthogonal signals in generated text. We believe this opens a complementary research direction to existing approaches, with practical extensions (multi-bit capacity, vector optimization, broader robustness testing) as natural next steps.
>
> - **Q3: Cross-evaluation between datasets and models:**
>
> We address this question in detail in our response Q2 to Reviewer 2 (bkYD), where we report cross-distribution experiments along both domain and architecture axes. In short, text-level F1 remains strong (80–94%) across dataset and fine-tuning shifts, it drops when using an other architecture for generation (Mistral → Llama). We will add the full results table to the appendix.

---

> > ### Author Rebuttal · Reviewer_TZxN · 2026-04-04
> >
> > Thank you for the rebuttal.
> >
> > **Q1:** Resolved. Thank you for clarifying that self-recognition emerges in base models and not just instruction-tuned models — this addresses my novelty concern relative to Ackerman & Panickssery.
> >
> > **Q2:** The additional MMLU results and SynthID comparison, and additional results in other rebuttles are appreciated and help contextualize the method's quality-detectability tradeoff.
> >
> > I will raise my score accordingly.

---

> > > ### Author Response · Authors · 2026-04-04
> > >
> > > Thank you for raising important points during the review, they prompted clarifications and updates that genuinely strengthened the paper. We are glad our responses and revisions met your expectations.
> > >
> > > You mentioned you would raise your score, we kindly invite you to update it in the system when you have a moment, as it does not appear to be reflected yet.
> > >
> > > We sincerely appreciate the time and effort you invested in this review.

---

### Official Review · Reviewer_YMww · 2026-03-08

**Soundness:** 3
**Presentation:** 3
**Significance:** 3
**Originality:** 3
**Overall Recommendation:** 4
**Confidence:** 4

**Summary:**

This paper investigates whether large language models (LLMs) encode self-recognition signals within their internal activations and proposes amplifying these signals by injecting a sparse random steering vector during generation. By embedding a signature directly into the activation space, the method enables high-accuracy attribution via lightweight probes or training-free cosine similarity without modifying the token-selection process. Empirically, the authors report near-perfect self-recognition on XL-Sum and reliable multi-identity attribution on ELI5 and Fresh News, demonstrating superior robustness to paraphrasing compared to KGW-style watermarks and a favorable sparsity-quality trade-off. While the mechanism is elegant and the results are compelling, the requirement for white-box access to internal activations at detection time limits its applicability to controlled environments rather than open-world watermarking. Consequently, while the submission addresses an important concept with a memorable and technically sound approach, its impact is currently constrained by the white-box assumption, an underdeveloped security model, and a need for broader baseline comparisons to fully contextualize its performance margins.

**Compliance With Llm Reviewing Policy:**

Affirmed.

**Final Justification:**

My final recommendation is Weak Accept. I found the paper to be original and technically interesting, especially in its use of sparse activation steering as a watermarking mechanism and in the evidence that the steering signal can be recovered from activations with strong attribution performance. The work is clearly written and empirically promising, and the rebuttal substantially strengthened my confidence in the submission by directly addressing my main concerns with additional clarification and experiments, including the quality-score formula, stronger confound controls, a re-evaluation of multi-class scaling under fixed per-class sample sizes, discussion of model mismatch, and more detail on parameter selection and multi-bit feasibility. I still view the white-box detection requirement, the security assumptions, and the limited breadth of baseline comparisons as meaningful limitations that constrain the current significance and practical scope of the method. However, after considering both the paper and the rebuttal, I believe the contribution is sound, novel, and sufficiently well supported to merit acceptance, and the rebuttal changed my evaluation by resolving enough of my earlier concerns to move me from weak reject to weak accept.

**Key Questions For Authors:**

1. **Quality-Score Mapping Formula:** Could you explicitly provide the formula used to map the DeBERTa classifier outputs to the $[0, 1]$ range in Appendix F? A mathematically sound implementation to resolve this would be the expected value under ordinal weights: $quality = 1.0 \cdot P(High) + 0.5 \cdot P(Medium) + 0.0 \cdot P(Low)$. Specifying your exact approach would greatly improve reproducibility.
2. **Stronger Baselines:** Could you evaluate the method against stronger, modern baselines (e.g., curvature-based detection, RepreGuard, or SAEMark) to better contextualize the performance margins you are reporting?
3. **Confound Controls in Summarization:** How resilient is the near-perfect no-prompt self-recognition accuracy to aggressive normalization of superficial artifacts, such as enforcing strict length matching, removing punctuation, or stripping markdown formatting?
4. **Model Mismatch:** Does the zero-shot cosine retrieval signal survive if the detector is a slightly different checkpoint than the generator—for instance, a quantized variant, a pruned model, or a slightly updated fine-tune?
5. **Multi-Class Scaling Re-evaluation:** Could you rerun the multi-class scaling experiment with a fixed number of samples *per class* to cleanly isolate the intrinsic separability of the vectors from the sample-budget confound?
6. **Robustness Under Attack:** Beyond the DIPPER-XXL paraphrasing evaluated in the paper , how does the method perform against broader adversarial attacks like translation chains, OCR-like text corruption, or mixed-source splicing?
7. **Multi-Bit Superposition:** You hypothesize that superimposing multiple sparse vectors with disjoint supports could create a high-capacity multi-bit watermark. Do you have preliminary empirical results proving that linear superposition is actually recoverable in practice without destructive interference?
8. **Parameter Selection Heuristics:** Can you provide a reproducible heuristic for how the extraction layer was optimized independently for each model, and discuss how the specific sparse support dimensions were chosen?

**Limitations:**

yes

**Strengths And Weaknesses:**

### Strengths

*
**Technical Novelty:** Shifting the watermarking paradigm from output-space token manipulation to activation-space steering is a remarkably clean and innovative approach.


*
**Optimal Steering Trade-offs:** The decision to use sparse random vectors (specifically, 99.7% sparsity) rather than dense vectors is well-motivated. The paper successfully demonstrates empirically that sparsity offers a vastly superior trade-off, maintaining high detectability while minimizing unwanted semantic perturbations and preserving output quality.


*
**Zero-Shot Retrieval:** One of the most intriguing findings is that a sparse steering signal survives the discretization step of token sampling and re-embedding. This directional alignment allows the signature to be retrieved via simple, training-free cosine similarity, proving the signal is geometrically meaningful and not just a statistical artifact.


*
**Experimental Rigor and Clarity:** The evaluation is commendably broad, spanning multiple model families including Llama-3 (8B, 3B, 1B) and Ministral-3-8B across both low- and high-entropy generation settings. The paper is also exceptionally well-structured; visual aids like Figure 1 clearly map the steering and retrieval workflow, while Figures 3 and 4 effectively communicate robustness and quality trade-offs.



### Weaknesses

*
**Practical Limitations:** The white-box assumption strictly limits deployment scenarios. Detection requires a forward pass through the exact model to encode the text into activations, which the authors note incurs a computational cost equivalent to 21% of auto-regressive generation time. This makes the framing of the method as a general-purpose watermarking solution somewhat overstated.


*
**Security Vulnerabilities:** The security model relies entirely on the secrecy of the steering configuration. As the authors acknowledge, if the specific steering vector and target layer are discovered or leaked, an attacker could trivially spoof the watermark.


*
**Narrow Baselines and Confounded Scaling:** Perplexity is utilized as the primary baseline for self-recognition , but the evaluation lacks comparisons against more modern contrastive detectors or recent activation-based watermarking methods like RepreGuard or SAEMark. Additionally, the multi-class scaling experiment holds the total training sample size constant as class counts grow, meaning the observed drop in accuracy is heavily confounded by a shrinking per-class sample budget.


*
**Opaque Quality-Score Mapping:** To evaluate text degradation, the authors rely on the external `quality-classifier-deberta` model. The paper states it extracts the softmax probability distribution for "High," "Medium," and "Low" quality tiers to derive a continuous score in the range of $[0, 1]$. However, the exact mathematical mapping formula is entirely omitted from Appendix F, hindering perfect reproducibility and obscuring how the tiers are weighted.

---

> ### Author Rebuttal · Authors · 2026-03-31
>
> We thank the reviewer for the careful reading and the encouraging assessment of our work.
>
> Q1. **Quality-Score Formula:**
>
> The formula proposed by the reviewer is indeed the one we use:
> $\hat{q} = 0 \cdot p_{Low} + 0.5 \cdot p_{Medium} + 1 \cdot p_{High}$. We will add this explicitly to Appendix F in the revision.
>
>
> Q2. **Stronger Baselines:**
>
> We agree that stronger baselines would help contextualize the potential of activation space for AI-GTD. We refer the reviewer to our response Q2 to Reviewer 4 (TZxN), where we evaluate robustness against SynthID and explain why fair comparison with SAEMark and SemStamp was not feasible, and to our response Q1 to Reviewer 2 (bkYD) where we discuss the Text Fluoroscopy baseline.
>
>
> Q3. **Confound Controls:**
>
> We ran additional experiments with even stronger confound controls: texts have been converted to be lowercase-only, all punctuation has been removed, and only pairs of human-written and LLM-generated summaries have been preserved where the lengths of the two summaries where in the closest 10% (<22 characters apart). This was done separately for six different XLSum subsets. Compared to baselines without the additional confound controls, classification accuracies for each domain changed by -0.2pp (all), +0.31pp (business), -0.41pp (entertainment), -0.22pp (technology), -0.24pp (uk), -0.47pp (world) (pp: percentage points). Additionally, classification performance in regions where summary lengths overlap is still excellent. These observations are further evidence supporting our finding that classification is not strongly confounded by formatting.  Further, texts are usually only 1-2 sentences long with low to moderate entropy, which makes confounds unlikely.
>
> Q4. **Model Mismatch:**
>
> We address this question in detail in our response Q2 to Reviewer 2 (bkYD), where we report cross-evaluation experiments along both dataset domain and architecture axes. In short, text-level F1 remains strong (80–94%) across dataset and model checkpoint shifts, as expected it drops when using an other architecture for generation (Mistral → Llama). The zero shot cosine similarity follows the same trend, persisting under similar model but degrading under a fully different one. We will add the full results table to the appendix.
>
> Q5. **Multi-Class Re-evaluation:**
>
> We reran the scaling experiment with a fixed number of samples per class, eliminating the data starvation confound. Results (token-level / text-level F1):
>
> | Number of Classes | 2 | 3 | 5 | 7 | 10 | 15 | 20 |
> | --- | --- | --- | --- | --- | --- | --- | --- |
> | Token Accuracy | 90% | 79% | 65% | 65% | 52% | 45% | 42% |
> | Sentence Accuracy | 99% | 95% | 88% | 90% | 80% | 71% | 69% |
>
> With a fixed number of samples per class, detection performance improves at larger class counts compared to the original experiment. There is further room for improvement through per-class hyperparameter tuning.
>
> Q6. **Robustness:**
>
> We selected DIPPER-XL with aggressive settings as our robustness benchmark because it is specifically designed to break watermarks and represents a strong upper bound on paraphrasing-based attacks. Broader robustness testing (covering translation chains, OCR-like corruption, or mixed-source splicing) would be informative for characterising failure cases and is a natural next step. As discussed in our introduction, our work is not intended as a ready-to-deploy watermarking system, but rather as a demonstration that activation space is a viable and robust channel for encoding recoverable signals, a substrate on which practical systems can be built.
>
> Q7. **Multi-Bit Superposition:**
>
> Preliminary results in the multi-bit setup with a small number of bits show that steering with multiple vectors on disjoint supports yields detection performance comparable to the multi-class setting on a per-bit basis. The main limitation is that superimposing multiple vectors densifies the aggregate steering signal, increasing quality degradation and weakening the detection-quality tradeoff. This can be mitigated through steering vector optimisation (see Q8 below) and preferring encodings with few active bits (minimising the Hamming weight).
>
> Q8. **Parameter Selection Heuristics:**
>
> The steering layer is selected by maximizing detection performance on a small calibration set. This yields a middle-layer optimum across all tested models, in line with observations from prior work on activation steering and probing.
>
> The sparsity pattern is sampled uniformly at random: a fixed fraction of vector dimensions are filled with values drawn from [-1, 1] and scaled by α. The sparcity level and α are selected by grid search to balance detectability and quality degradation. As discussed in Section 4, the choice of a steering vector and its support dimensions is not neutral with respect to this tradeoff, and could be further improved through selective vector sampling or a learned optimization procedure.

---

> > ### Author Rebuttal · Reviewer_YMww · 2026-04-02
> >
> > Thank you for the detailed rebuttal. The additional experiments and clarifications adequately address my main concerns, especially regarding confound controls, multi-class scaling, model mismatch, reproducibility of the quality metric, and the practicality of the proposed setup. While some limitations remain, I believe the authors have provided sufficient evidence and justification to support their claims, and I am satisfied that my concerns have been resolved. I will raise my score accordingly.

---

> > > ### Author Response · Authors · 2026-04-04
> > >
> > > Thank you for your thoughtful and constructive feedback. Your comments helped us clarify important points and strengthen the paper. We are glad that our additional experiments and responses addressed your concerns, and we will incorporate these clarifications and updates into the final version.
> > >
> > > We sincerely appreciate your time and detailed review.

---

### Official Review · Reviewer_bkYD · 2026-03-10

**Soundness:** 3
**Presentation:** 3
**Significance:** 3
**Originality:** 3
**Overall Recommendation:** 4
**Confidence:** 4

**Summary:**

This paper studies whether large language models (LLMs) encode self-recognition signals in their internal activations and proposes a simple inference-time “activation steering” mechanism that adds a sparse random vector at a chosen layer during generation to create a recoverable fingerprint. Using linear probes for self-recognition and an MLP or cosine-similarity for multi-model attribution, the authors report very high accuracy (often >98% AUROC/F1) across several Llama and Ministral models, including in prompt-agnostic, low-entropy summarization settings, and show improved robustness to paraphrasing relative to a token-level watermark baseline. They also analyze sparsity–quality trade-offs and show that steering vectors can be partially retrieved from activations via cosine similarity without training.

**Compliance With Llm Reviewing Policy:**

Affirmed.

**Key Questions For Authors:**

1. **Stronger baselines.** Can you include DetectGPT variants, binocular/cross-perplexity scoring, and at least one activation-based detector (RepreGuard or Text Fluoroscopy) under the same datasets and evaluation splits? How do these compare to your method in both the prompt-conditioned and prompt-agnostic regimes?
2. **Confound controls for self-recognition.** If you strictly match each human–model pair by length and remove formatting/punctuation artifacts before activation extraction, do the near-perfect AUROC scores materially change? This would help establish whether the signal reflects stylistic fingerprints rather than structural artifacts.
3. **Detector–generator mismatch.** What happens when detection uses a slightly different checkpoint (e.g., a quantized or pruned variant of the same base model, or a different fine-tune)? Does cosine-similarity retrieval survive cross-checkpoint or cross-architecture evaluation? This is directly relevant to practical deployment.
4. **Multi-class scaling with fixed per-class samples.** Can you re-run the scaling experiment (Figure 2) with a fixed number of samples per class, and provide learning curves to separate data scarcity from intrinsic signal separability? As currently reported, the declining F1 may primarily reflect data starvation.
5. **Multi-bit superposition experiment.** Is the steering signal linearly superimposable in practice? Even a small pilot experiment simultaneously steering with two disjoint sparse vectors and attempting to recover both independently would be informative for the proposed multi-bit watermark extension.

**Limitations:**

The paper includes a dedicated Limitations section and an Impact Statement that acknowledge white-box detection overhead, quality evaluator biases, and security risks arising from vector secrecy. These disclosures are appropriate and honest. However, the discussion of robustness to adversarial attacks beyond paraphrasing, and the privacy implications of linking text to specific deployments, could be expanded. No substantive ethical concerns are raised that would require additional review.

**Strengths And Weaknesses:**

**Strengths**
1. Technical novelty. The method uses random, sparse activation-space steering at generation time to embed a detectable signal without modifying token-selection logic or requiring specialized watermark sampling routines. The training-free retrieval via cosine similarity between steering vectors and re-encoded activations is an elegant result suggesting the signal survives the discrete sampling–re-embedding cycle. Positioning "self-recognition" through internal activations as a distinct axis from perplexity-based or token-level watermarks, with potential for multi-bit encoding via superposition of sparse vectors, is conceptually fresh.
2. Experimental breadth. The evaluation spans multiple model families and sizes (Llama-3.x at 1B/3B/8B and Ministral-3-8B), three datasets (XL-Sum, ELI5, Fresh News), and both prompt-conditioned and prompt-agnostic settings. The paper includes ablations on vector sparsity vs. dense steering, performance vs. number of steered identities, token-position dynamics, and detection granularity (token vs. majority-vote text aggregation). OOD domain checks for self-recognition and robustness tests with strong paraphrasing (DIPPER-XXL) are also included.
3. Clarity and reproducibility. The method is clearly described with an explicit threat model, activation aggregation protocol, and detector training procedure. Informative figures and appendices cover layer-wise performance, token-position dynamics, paraphrase settings, prompts, and hyperparameters. Code and data are promised for reproduction.
4. Significance. The work addresses an important problem: provenance and attribution of LLM-generated text, moving beyond binary detection toward model-level and steered-variant attribution. It provides empirical evidence that internal representations can host non-semantic, quasi-orthogonal signals without obvious generation quality degradation, opening a promising avenue for activation-level watermarking.

**Weaknesses**
1. White-box constraint limits practical scope. Detection requires access to the same model's internal activations at attribution time. Practical generalization to cross-vendor attribution, black-box APIs, or open-set scenarios is not addressed. The security guarantee is entirely secrecy-dependent; no red-team analysis (e.g., vector estimation attacks) is provided.
2. Insufficient baseline comparisons. For self-recognition, only perplexity is used as a baseline. No comparison is included against strong modern detectors such as DetectGPT variants, binocular/cross-perplexity scoring, Text Fluoroscopy, or RepreGuard. For the watermarking component, only KGW is compared; recent robust watermarks such as SAEMark or SemStamp are absent.
3. Potential confounds in self-recognition. Near-perfect AUROCs in low-entropy summarization without prompts raise questions about whether the signal reflects genuine stylistic fingerprints or residual confounds (length distribution, formatting patterns, model-specific structural priors). Length and domain controls are partially addressed, but per-sample length matching, formatting normalization, and shuffled baseline tests are not reported.
4. Multi-class scaling conflates difficulty with data scarcity. The experiment fixing total sample count across class conditions makes it impossible to distinguish intrinsic signal separability from data starvation as the number of identities grows. No per-class fixed-sample or learning-curve analysis is provided.
5. Limited robustness and quality evaluation. Robustness is tested only via DIPPER paraphrasing; translation chains, formatting degradation (e.g., OCR noise), partial human edits, copy-paste splicing, and variable temperature/decoding parameters are not explored. Quality assessment relies on a single external DeBERTa-based classifier with known formatting biases; no human evaluation or task-level metrics (e.g., ROUGE on XL-Sum) are reported.
6. Layer selection underdocumented. The rationale for choosing the steering/extraction layer is only briefly described as "middle layers work best"; no systematic heuristic or sensitivity analysis sufficient for deployment guidance is provided.

---

> ### Author Rebuttal · Authors · 2026-03-31
>
> We thank the reviewer for the thorough and constructive feedback.
>
> Q1. **Stronger baselines.**
>
> Our method targets activation-space self-recognition as well as leveraging this space as a substrate for watermark, not general AI-generated text detection, therefore binary detectors such as *DetectGPT* (Mitchell et al. 2023) are not directly comparable to our setting. To address the reviewer's concern, we instead evaluate whether residual stream patterns can distinguish texts from different LLMs, analogous in spirit to *Text Fluoroscopy* (Yu et al.,2024). The table below reports cross-model classification accuracy, with in-distribution performance on the diagonal.
>
> | ⬇️ Train \ Test ➡️ | Ministral-3-8B | Llama-3.1-8B | Llama-3.2-3B | Llama-3.2-1B |
> | --- | --- | --- | --- | --- |
> | Ministral-3-8B | 0.9986 | 0.8089 | 0.8068 | 0.7709 |
> | Llama-3.1-8B | 0.9865 | 0.9800 | 0.9633 | 0.9354 |
> | Llama-3.2-3B | 0.9860 | 0.9645 | 0.9798 | 0.9428 |
> | Llama-3.2-1B | 0.9807 | 0.9493 | 0.9510 | 0.9772 |
>
> Classifiers trained on Ministral-3-8B activations degrade sharply when evaluated on activations induced by Llama-family texts. The reverse does not hold: Llama activation patterns appear relatively consistent regardless of the generating model. We will incorporate these findings into the final version.
>
> We also conducted additional comparaison to SythID descirbed in response Q2 to Reviewer 4 (TZxN)
>
> Q2. **Confound controls for self-recognition.**
>
> We ran additional confound control experiments described in Q.3 of Reviwer 3 (YMww). In short, performances are not affected by structural artifacts.
>
> Q3. **Cross-model evaluation:**
>
> To assess out-of-distribution robustness, we ran cross-evaluation experiments along two axes. With the training pipeline noted as:
>
> $Train: Prompt_1 \in D_1 \rightarrow LLM_a \rightarrow Text_{a1} \rightarrow LLM_a \rightarrow A_{a1} \rightarrow MLP_a$
>
> *Domain adaptation* tests on a different dataset with the same model:
>
> $Test:  Prompt_2 \in D_2 \rightarrow LLM_{a} \rightarrow Text_{a2} \rightarrow LLM_{a} \rightarrow A_{a2} \rightarrow MLP_{a}$
>
> *Model adaptation* tests on a different generating LLM, with activations extracted from the original training model:
>
> $Test:  Prompt_1 \in D_1 \rightarrow LLM_{b} \rightarrow Text_{b1} \rightarrow LLM_{a} \rightarrow A_{b\to a2} \rightarrow MLP_{a}$
>
> In both cases, activation collection and MLP inference always use the training setup, reflecting a realistic deployment scenario where the provider controls the detection pipeline. We report the token accuracy and the aggregated text accuracy.
>
> | Train setup \ Test setup | Llama3.1-8B-Instruct + ELI5 | Llama3.1-8B + News | Llama3.1-8B-Instruct + News | Llama3.1-8B + ELI5 | Ministral3-8B-Base + News | Ministral3-8B-Base + ELI5 |
> | --- | --- | --- | --- | --- | --- | --- |
> | Llama3.1-8B-Instruct + ELI5 (B) | 90% / 99% | 74% / 80% | 77% / 84% | 74% / 85 % | 55% / 52% | 58% / 51 % |
> | Llama3.1-8B + News (A) | 76 % / 84 % | 90% / 99% | 86% / 94 % | 78% / 90% | 56% / 61 % | 61 % / 65% |
>
> As expected, out-of-distribution performance degrades, though models trained on higher-entropy data generalize better across both domain and architecture shifts, likely due to richer activation variability during training. Notably, the watermark signal partially survives even cross-architecture re-encoding (Mistral → Llama), suggesting the steering leaves traces beyond the generating model. Training on a diverse mixture of prompt types and model variants is a natural path to a more robust detector.
>
> Zero-shot cosine similarity degrades moderately under checkpoint shifts within the same family, and drops to chance level with architecture change, confirming that the geometric alignment of the steering signal is architecture-dependent and that this retrieval mode is best suited to same-architecture deployment.
>
> Q4. **Multi-class:**
>
> The experiment has been ran and is exposed in Q5. of Reviewer 3 (YMww). In short, the detection performance for a larger number of classes got improved, suggesting that there was data starvation.
>
> Q5. **Multi-bit:**
>
> Some preliminary results are explained in Q7. of Reviewer 3 (YMww). In short, superimposing multiple sparse vectors densifies the aggregate steering signal and degrades generation quality, making vector optimisation a necessary prior step before this extension can be fully evaluated.
>
> 6. **Robustness and quality:**
>
> We discuss our robustness benchmark choice in our response Q6 to Reviewer3(YMww) and provide additional quality assessment results on MMLU in our response Q2 of Reviewer4(TZxN)
>
> 7. **Layer choice:**
>
> We describe our parameter selection procedure in our response Q8 to Reviewer3(YMww) and will make these choices more explicit in the revision

---

> > ### Author Rebuttal · Reviewer_bkYD · 2026-04-04
> >
> > I acknowledge the authors' rebuttal and appreciate the additional experiments, particularly the cross-model evaluation table (Q3), which honestly characterizes the method's limitations under architecture shift, and the clarification that binary AI-text detectors are not the right comparison target (Q1).
> > However, several concerns remain partially open:
> >
> > - Baselines (Q1): The cross-model table is informative, but the absence of any activation-based detector comparison (e.g., RepreGuard) leaves the positioning of the method relative to existing activation-level approaches unclear. The asymmetry in the cross-model table (Ministral probes failing on Llama but not vice versa) also warrants discussion.
> > - Confound controls and multi-class scaling (Q2, Q4): These are deferred to other reviewers' threads. I would appreciate the relevant details being consolidated in the revised paper so reviewers and readers can evaluate them directly.
> > - Multi-bit superposition (Q5): The honest acknowledgment that superposition degrades quality is appreciated, but it means the multi-bit watermark claim in the paper remains speculative. The revised version should temper this framing accordingly.
> > - Robustness (original weakness 5): Robustness beyond DIPPER paraphrasing remains unaddressed (translation chains, partial edits, OCR noise, etc.). While these may be hard to fully explore in a rebuttal, the paper would benefit from at least acknowledging these gaps more explicitly.

---

> > > ### Author Response · Authors · 2026-04-04
> > >
> > > Thank you for your careful assessment and thoughtful feedback. We are currently revising the paper to incorporate these points more clearly, including the positioning of our method relative to activation-based approaches, the discussion of cross-model asymmetry, the clarifications on confound controls and multi-class scaling, and a more explicit presentation of the current limitations regarding multi-bit watermarking and robustness.
> > >
> > > We sincerely appreciate the time and care you devoted to reviewing our work.

---

### Official Review · Reviewer_LHmB · 2026-03-13

**Soundness:** 3
**Presentation:** 2
**Significance:** 3
**Originality:** 3
**Overall Recommendation:** 5
**Confidence:** 4

**Summary:**

The paper aims to propose a new method to attribute text to one of many LLMs. During generation, a fixed sparse random vector is added to the model's activations at a chosen layer at every token step. This consistently cahnges the generation trajectory, imprinting a signal into the token choices. At detection time, the text is fed back through the same model without steering, activations are extracted at the same layer, and the original steering vector is recovered either via a trained MLP or simple cosine similarity. The vector is kept very sparse to avoid quality degradation, and its randomness makes it roughly orthogonal to the model's semantic directions.

**Compliance With Llm Reviewing Policy:**

Affirmed.

**Final Justification:**

The author response, particularly the new results, addressed my concerns.

**Key Questions For Authors:**

none

**Limitations:**

yes

**Strengths And Weaknesses:**

The paper addresses a practical and timely problem, and the proposed method is refreshingly simple. The observations that sparse random vectors can encode a recoverable signal without degrading generation quality is compelling and well-supported by the quality-detectability tradeoff analysis in the paepr. The self-recognition results are strong, particularly in the prompt-agnostic setting where the perplexity baseline collapses. The cosine similarity recovery experiment (Section 3.5) is especially interesting because it demonstrates that the signal survives the discretization bottleneck of token sampling without any trained detector, lending credibility to the claim that the signal is encoded in the text rather than being a classifier artifact.

The two main contributions feel somewhat disconnected: self-recognition and steering-based attribution solve different problems, yet the paper frames one as an amplification of the other without making that bridge convincing. The white-box access requirement at detection time is a significant practical limitation that undermines comparisons with watermarking methods like KGW, which only need the secret key and the text. The evaluation is narrow in several respects: only KGW is compared against, and the quality evaluation relies on a single external classifier that the authors themselves acknowledge is biased. The scalability analysis (Figure 2) shows clear degradation beyond a handful of classes, and the proposed combinatorial multi-bit extension is not supported.

---

> ### Author Rebuttal · Authors · 2026-03-31
>
> We thank the reviewer for the thorough and constructive feedback.
>
> 1. **Framing steering-based attribution as “amplification” of self-recognition**.
>
> We agree that "amplification" is misleading and will revise the terminology. The two contributions share a common foundation: both detect signals from the residual stream, one natural and one artificially injected, but serve distinct goals. Rather than amplifying a pre-existing self-recognition signal, the steering method imprints an individualised signature into the activation space, enabling multi-model attribution beyond binary self/non-self recognition. The logical connection between the two parts is therefore that steering-based attribution operates in a strictly more challenging setting than self-recognition, substantiating the broader claim that residual stream patterns are a reliable substrate for provenance detection.
>
> 2. **KGW as baseline for watermarking schemes.**
>
> While additional baseline comparisons would be informative, KGW is the standard reference against which virtually every proposed watermarking scheme is benchmarked, with well-established performance in generation setting such as Fresh News. We nonetheless conducted additional comparisons, discussed in our response Q1 to Reviewer 2 (bkYD) and Q2 to Reviewer 4 (TZxN), which we will incorporate into the final version. More broadly, our work is not intended as a ready-to-deploy watermarking system, but as a demonstration that activation space is a viable and robust channel for encoding recoverable signals, a substrate on which practical systems can be built.
>
> 3. **Bias in quality assessment**.
>
> To address potential biases in our quality assessment, we ran additional capability evaluations on MMLU, revealing a consistent yet small performance degradation under steering. Full details are provided in our response Q2 to Reviewer 4 (TZxN) and will be added to revision.
>
> 4. **Missing support for the multi-bit extension**.
>
> Preliminary experiments confirm that independently injected sparse vectors can be recovered from multi-bit steered generations. However, superimposing multiple vectors densifies the aggregate steering signal, increasing quality degradation. A viable multi-bit extension therefore requires steering vector optimisation (to improve the detectability-quality tradeoff) as a necessary prior step, which we identify as the most natural direction for future work.
>
> 5. **White-box access requirement limiting practicality and comparability to other methods.**
>
> As discussed in our limitations section, the main practical constraint of our approach is the computational overhead at detection time, which requires a forward pass through the model. Since the extraction layer is consistently the middle layer of the network, this overhead could be approximately half that of a full forward pass and negligible at insertion time. This makes our approach more practical for deployment by black-box LLM providers than resampling-based methods, which incur substantial generation-time overhead. Detection could additionally be exposed via an API by the model provider, preserving black-box behaviour for end users in the same way as existing schemes. The method also applies naturally to settings where white-box access is standard, such as training data filtering. We therefore argue that the practical gap between white-box and black-box watermarking is narrower than it may initially appear.

---

> > ### Author Rebuttal · Reviewer_LHmB · 2026-04-03
> >
> > Thank you, your response addresses my concerns and I will raise my score to "accept".

---

> > > ### Author Response · Authors · 2026-04-04
> > >
> > > Thank you for your careful review and reassessment. We are glad our response addressed your concerns, and we sincerely appreciate your time, effort, and interest in our method.

---

### Decision · Program_Chairs · 2026-04-30

**Decision:**

Accept (regular)

**Comment:**

This paper investigates whether large language models encode self-recognition signals in their internal activations and proposes a new method to attribute text to specific models using activation steering. The authors introduce an inference-time mechanism that adds a fixed sparse random vector to the model's activations at a chosen layer during generation. This injects a recoverable signature into the token choices. At detection time, the text is passed through the same model to extract activations, allowing the original steering vector to be recovered using a trained MLP or training-free cosine similarity. The method keeps the vector sparse to prevent quality degradation and random to remain roughly orthogonal to semantic directions. Empirical evaluations demonstrate multi-model attribution accuracy often exceeding 98% AUROC/F1, superior robustness to paraphrasing compared to KGW-style watermarks, and a favorable sparsity to quality trade-off. The paper additionally studies self-recognition capabilities, showing that probes trained on model activations can effectively distinguish between human and LLM-generated text.

This paper received all positive scores from the reviewers, who found the work to be original, technically interesting, and empirically promising. During the rebuttal phase, three reviewers raised their scores to recommend acceptance. These score increases were primarily due to the authors providing additional experiments and clarifications that successfully addressed initial concerns regarding confound controls, multi-class scaling, evaluation under architecture shifts, and the novelty of the self-recognition results compared to prior work. While some practical limitations regarding the white-box detection requirement were noted, the consensus is that the rebuttal effectively resolved the primary issues. Given the sound contribution and novel approach to activation steering, the paper merits acceptance.